# Computation and Memory-Efficient Model Compression with Gradient Reweighting

**Zhiwei Li**[1][*]**, Yuesen Liao**[1][*]**, Binrui Wu**[1]**, Yuquan Zhou**[1]
**Xupeng Shi**[2]**, Dongsheng Jiang**[3]**, Yin Li**[3]**, Weizhong Zhang**[1,4][†]
[1]Fudan University, [2]Northeastern University, [3]Huawei Inc.
[4]Shanghai Key Laboratory of Intelligent Information Processing
{zwli23, ysliao24, brwu23, yqzhou21}@m.fudan.edu.cn
dongsheng_jiang@outlook.com, liyin9@huawei.com
weizhongzhang@fudan.edu.cn

## Abstract

Pruning is a commonly employed technique for deep neural networks (DNNs) aiming at compressing the model size to reduce computational and memory costs during inference. In contrast to conventional neural networks, large language models (LLMs) pose a unique challenge regarding pruning efficiency due to their substantial computational and memory demands. Existing methods, particularly optimization-based ones, often require considerable computational resources in gradient estimation because they cannot effectively leverage weight sparsity of the intermediate pruned network to lower compuation and memory costs in each iteration. The fundamental challenge lies in the need to frequently instantiate intermediate pruned sub-models to achieve these savings, a task that becomes infeasible even for moderately sized neural networks. To this end, this paper proposes a novel pruning method for DNNs that is both computationally and memory-efficient. Our key idea is to develop an effective reweighting mechanism that enables us to estimate the gradient of the pruned network in current iteration via reweigting the gradient estimated on an outdated intermediate sub-model instantiated at an earlier stage, thereby significantly reducing model instantiation frequency. We further develop a series of techniques, e.g., clipping and preconditioning matrix, to reduce the variance of gradient estimation and stabilize the optimization process. We conducted extensive experimental validation across various domains. Our approach achieves 50% sparsity and a $1.58\times$ speedup in forward pass on Llama2-7B model with only 6 GB of memory usage, outperforming state-of-the-art methods with respect to both perplexity and zero-shot performance. As a by-product, our method is highly suited for distributed sparse training and can achieve a $2 \times$ speedup over the dense distributed baselines.

## 1 Introduction

The explosive growth in the scale of deep neural networks (DNNs), especially large language models (LLMs) [1, 6, 48], has introduced practical challenges, including computational expense, memory usage, and latency in inference. Model pruning [13, 14, 21, 28, 34], particularly structured pruning, has emerged as a promising solution, aiming to obtain a sparse neural network by removing redundant components. For conventional DNNs, it has been reported that existing methods [8, 23] can enhance

---

[*]Equal Contribution.
[†]Corresponding Author.

the inference efficiency and reduce the memory consumption by orders of magnitudes, with only a slight performance degradation, facilitating the deployment of DNNs on low-power devices.

In contrast to conventional DNNs, LLMs have substantial computational and memory demands. The users in downstream applications always have significantly less computational and data resources for pruning compared to those used for training. Consequently, LLMs pose a unique challenge for pruning. Metric-based pruning methods [2, 32, 55] are proposed to overcome this challenge; however, their manually designed rules can adversely affect the performance of the pruned model, which is more evident at higher sparsity levels. Moreover, due to the differing parameter redundancy across layers, manually setting the pruning rate for each layer is almost impossible. Thus, these methods struggle to achieve global heterogeneous pruning and are often forced to apply uniform pruning across layers [3, 46]. Optimization-based pruning methods [18, 53], which better preserve model performance, typically rely on training with backpropagation, making them difficult to apply in resource-constrained applications [31, 57].

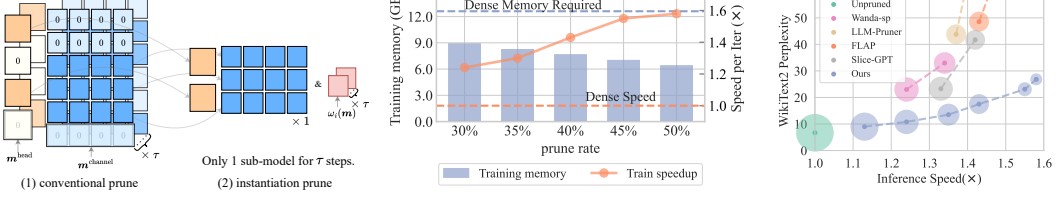

(a) Algorithm Overview Diagram.  (b) Memory Cost & Inference Speed.  (c) PPL vs. Inference Speed.

Figure 1: (a) In the case of $\tau$ iterations, vanila implementation need to instantiate sub-model $\tau$ times. In contrast, with the reweighting technique, a single sub-model can be used continuously for $\tau$ steps. This is a schematic; in practice, multiple sub-models can be sampled (see Algorithm 1). (b) With our method, higher prune rates lead to lower training memory cost and faster forward computation. Notably, with 50% prune rate, we prune Llama2-7B (12.6 GB in BF16) using only **6.3 GB** memory, while achieving **1.58** $\times$ speedup in the forward pass. (c) Circle size is proportional to pruned model's memory usage. We achieve substantial inference speedup while maintaining a low perplexity.

We argue that leveraging the sparse structure of the model is a natural idea to develop computationally and memory-efficient pruning algorithms. This can be implemented by instantiating the sparse subnetwork during pruning. However, as the subnetwork structure is updated in each iteration during pruning, the model instantiation needs to be performed frequently, which could leads to unbearable time cost. To this end, we propose a computationally and memory-efficient optimization-based pruning algorithm with a novel reweighting technique to reduce the model instantiation frequency. This approach effectively leverages the sparsity by completely discarding pruned parameters, preventing them from participating in computation and storage. Our carefully designed reweighting mechanism enables us to estimate the gradient of the pruned network in current iteration via simply reweighting the gradient estimated on an outdated intermediate submodel instantiated at an earlier stage, in this way the model instantiation frequency can be reduced. Our algorithm is illustratively shown in Fig.1a. We further integrate our reweighting technique with policy gradient estimator to bypass the computationally expensive backpropagation to update the sparse sub-model structural parameters. Finally, considering the additional variance introduced by reweighting and policy gradient estimator, we develop a series of variance reduction techniques to stabilize training, including *clipping* and *preconditioning matrix*, which are detailed in Section 4.2.

Extensive experiments on LLM demonstrate that our method reduces memory requirements during the pruning process and provides substantial computational speedup, particularly under high pruning rate. On the Llama2-7B model, our algorithm achieved 50% sparsity with only 6 GB of memory usage and delivered a 1.58 $\times$ speedup in forward pass, as shown in Fig. 1b, while maintaining state-of-the-art performance in terms of perplexity and zero-shot capabilities. In Fig. 1c, we present a visual comparison of the model's speed and perplexity after pruning with different methods.

Additionally, our algorithm is highly compatible with distributed sparse training [4]. In this setting, we can send the instantiated sub-models to the respective clients. With the reweighting technique, each client can work on its sub-model for a long period, reducing the frequency of model weights transmission. During training, by passing only two real numbers, i.e., the reweighting value and

the training loss, we can provide clients with essential structural information and update structural parameters on the server based on the latest client results in real time.

Our main contributions can be summarized as follow:

- We propose a novel pruning approach that, for the first time, instantiates intermediate pruned sub-models throughout the pruning process. This technique effectively reduces computational and memory costs by limiting the frequency of instantiation, thereby enhancing overall efficiency in pruning.

- Our reweighting technique enables unbiased policy gradient estimation based on outdated structural information and reduces instantiation frequency. In addition, we design variance reduction techniques to stabilize the pruning process.

- Our approach achieves 50% sparsity and a $1.58\times$ speedup in forward pass on Llama2-7B with only 6 GB of memory usage, outperforming state-of-the-art methods with respect to both perplexity and zero-shot performance.

- As a byproduct, our algorithm is inherently well-suited for distributed sparse training, reducing communication overhead during training process.

## 2 Related Work

### 2.1 Conventional Neural Network Pruning

Neural network pruning [20] is a technique aimed at reducing model parameters, focusing on reducing storage costs and inference time. Conventional pruning algorithms [13, 19, 29, 38, 51] generally operate on traditional network, establishing criteria (e.g. weight magnitude, importance score) to assess and prune the parameters or modules, while minimizing performance degradation. These pruning methods make the deployment of some neural networks on resource-limited devices possible.

### 2.2 LLM Pruning

For LLMs, due to the large number of parameters and the resource mismatch between training and pruning, the discussed conventional pruning algorithms are often difficult to apply. An increasing variety of pruning algorithms are being designed specifically for LLMs, which can be divided into two main types based on whether structural parameters optimization is involved: metric-based [46, 49, 60, 63, 65] and optimization-based [7, 17, 27, 31].

**Metric-Based Pruning.** Metric-based pruning is similar to traditional methods mentioned above but is tailored to accommodate specific characteristics of LLMs, such as large number of parameters and the outlier activation in certain channels [11, 59]. These metrics are often more meticulously designed: for instance, Wanda [46] incorporates activations into the importance score; SparseGPT [15] efficiently calculates Hessian's inverse to set up metrics and parameter reconstruction; LLM-Pruner [34] groups the module by graph information, and deletes them together after obtaining the scores; FLAP [2] extends the metric globally through score normalization. These methods are typically limited to layers or matrix, and their performance often deteriorates under high sparsity.

**Optimization-Based Pruning.** By training the structural parameters, the optimization-based pruning algorithm can learn a more effective sparse structure. For instance, [54, 57] employs reparameterization to learn masks, pruning the model down to a fixed size, [57] additionally continues pretraining to restore model capability; APT [66] introduces a dynamic low-rank structure similar to LoRA [25], allowing adaptive fine-tuning and pruning. NutePrune [31] employs the model with lower sparsity as a teacher for distillation during pruning. Overall, optimization-based LLM pruning methods typically require gradient computation through the chain rule, which inevitably demands higher computational costs, making practical application more challenging.

Most LLM pruning methods fail to leverage sparse structures, and the resource mismatch between training and pruning calls for a computationally and memory-efficient algorithm.

# 3 Preliminary

**Framework of Probabilistic Masking for Pruning.** We adopt the sparse training framework for neural networks proposed in [67]. Given a neural network parameterized by $\boldsymbol{W} = \{\boldsymbol{w}_i\}_{i=1}^K$, where $K$ is the number of modules and $\boldsymbol{w}_i$ represents the parameter matrix of a module within the network, e.g., a head in Attention, or a channel in CNNs. Let $\boldsymbol{m} = [\boldsymbol{m}_1, \ldots, \boldsymbol{m}_K]$ denotes the corresponding binary mask for $\boldsymbol{W}$. The module $\boldsymbol{w}_i$ is pruned when $\boldsymbol{m}_i = 0$ and retained when $\boldsymbol{m}_i = 1$. By further reparameterizing the mask $\boldsymbol{m}$ with `Bernoulli` random variabels, i.e., $\boldsymbol{m}_i \sim \text{Bernoulli}(\boldsymbol{s}_i)$ with the probability of $\boldsymbol{s}_i$ to be 1, the structured pruning framework can be formalized as follows:

$$
\min_{\boldsymbol{s}} \mathbb{E}_{p(\boldsymbol{m}|\boldsymbol{s})} \mathcal{L}(\boldsymbol{W} \circ \boldsymbol{m}) \triangleq \frac{1}{N} \sum_{i=1}^N \ell(f(\boldsymbol{x}_i; \boldsymbol{W} \circ \boldsymbol{m}), \boldsymbol{y}_i),
$$
$$
s.t. \ \|\boldsymbol{s}\|_1 \leq (1-\rho)K \text{ and } \boldsymbol{s} \in [0,1]^K,
$$

(1)

where $\{(\boldsymbol{x}_i, \boldsymbol{y}_i)\}_{i=1}^N$ are the training samples, $\rho$ denotes the prune rate, $f(\cdot; \boldsymbol{W} \circ \boldsymbol{m})$ represents the sparse neural network, $\ell(\cdot, \cdot)$ is the loss function. Here, the optimization of $\boldsymbol{W}$ is omitted as we focus on pruning instead of training.

The base solver proposed in [67] optimizes above problem using policy gradient estimator, which enables unbiased gradient estimation of $\boldsymbol{s}$ using only forward propagation. After sampling a mini batch $\mathcal{B}$ and $\boldsymbol{m}$ from $p(\boldsymbol{m}|\boldsymbol{s})$, the specific update procedure for each iteration is as follows:

$$
\boldsymbol{s} \leftarrow \text{proj}_{\mathcal{S}}(\boldsymbol{s} - \eta \boldsymbol{g_s}), \text{ where } \boldsymbol{g_s} = \mathcal{L}_{\mathcal{B}}(\boldsymbol{m}) \nabla_{\boldsymbol{s}} \log(p(\boldsymbol{m}|\boldsymbol{s})),
$$

(2)

where $\mathcal{S} \triangleq \{\boldsymbol{s} : \|\boldsymbol{s}\|_1 \leq (1-\rho)K \text{ and } \boldsymbol{s} \in [0,1]^K\}$ is the feasible domain and $\text{proj}_{\mathcal{S}}(\cdot)$ is the projection operator[1] on $\mathcal{S}$. Notably, $\nabla_{\boldsymbol{s}} \log(p(\boldsymbol{m}|\boldsymbol{s})) = \frac{\boldsymbol{m}-\boldsymbol{s}}{\boldsymbol{s}(1-\boldsymbol{s})}$ can be computed explicitly and thus the entire process is more efficient compared to traditional backpropagation-based algorithms.

# 4 Method

In this section, we introduce our method in three parts: the reweighting technique to reduce the instantiation frequency in Section 4.1; the detailed framework of efficient pruning in Section 4.2; and finally, the extension to distributed sparse training in Section 4.3.

## 4.1 Reweighted Policy Gradient Estimator

**Sub-model Instantiation.** We first give the definition of instantiation as follows:

**Definition 4.1.** *Given the parameters $\boldsymbol{W}$ of a DNN and the corresponding masks $\boldsymbol{m}$, we **instantiate** a compact sub-model by creating a new neural network to delete the pruned modules, which can be written as $f(\cdot; \boldsymbol{W_m})$ with the parameters $\boldsymbol{W_m} = \{\boldsymbol{w}_i\}_{i \in \boldsymbol{I}}$ and $\boldsymbol{I} = \{i : \boldsymbol{m}_i = 1, i = 1, \ldots, K\}$.*

It can be seen that if we utilize $\boldsymbol{m}$ to instantiate a compact sub-model at each step, we can reduce memory consumption and accelerate forward speed. However, frequent instantiations lead to unbearable time costs. Considering this issue, we propose a new reweighting technique to reduce instantiation frequency, allowing to use outdated compact sub-model to optimize the structural parameters $\boldsymbol{s}$.

**Basic Idea.** Our reweighting technique is inspired by importance sampling, and its general form is:

$$
\mathbb{E}_{\mathbf{x} \sim p(\cdot)}[f(\mathbf{x})] = \int f(\mathbf{x})p(\mathbf{x})\mathrm{d}\mathbf{x} = \int q(\mathbf{x})f(\mathbf{x})\frac{p(\mathbf{x})}{q(\mathbf{x})}\mathrm{d}\mathbf{x} = \mathbb{E}_{\mathbf{x} \sim q(\cdot)}\left[f(\mathbf{x})\frac{p(\mathbf{x})}{q(\mathbf{x})}\right].
$$

(3)

This equation indicates that the expectation of $f(\mathbf{x})$ over distrbution $p(\cdot)$ equals to that of $f(\mathbf{x})\frac{p(\mathbf{x})}{q(\mathbf{x})}$ over $q(\cdot)$. The item $\frac{p(\mathbf{x})}{q(\mathbf{x})}$ is the importance weight. This means one can estimate the mean of a random variable in another probability space by choosing a proper distribution.

---

[1]The computation of the projection operator is detailed in Appendix D.

**Reweighting Technique Derivation.** Based on the above analysis, we develop our reweighted policy gradient estimator below. Firstly, we have

$$\nabla_{\boldsymbol{s}} \mathbb{E}_{p(\boldsymbol{m}|\boldsymbol{s})} \mathcal{L}(\boldsymbol{m})$$

$$= \nabla_{\boldsymbol{s}} \int p(\boldsymbol{m}|\boldsymbol{s}) \mathcal{L}(\boldsymbol{m}) \mathrm{d}\boldsymbol{m} = \int \mathcal{L}(\boldsymbol{m}) \nabla_{\boldsymbol{s}} p(\boldsymbol{m}|\boldsymbol{s}) + \underbrace{p(\boldsymbol{m}|\boldsymbol{s}) \nabla_{\boldsymbol{s}} \mathcal{L}(\boldsymbol{m})}_{=0} \mathrm{d}\boldsymbol{m}$$

$$= \int \mathcal{L}(\boldsymbol{m}) p(\boldsymbol{m}|\boldsymbol{s}) \nabla_{\boldsymbol{s}} \log(p(\boldsymbol{m}|\boldsymbol{s})) \mathrm{d}\boldsymbol{m} \stackrel{(3)}{=} \int \mathcal{L}(\boldsymbol{m}) \nabla_{\boldsymbol{s}} \log(p(\boldsymbol{m}|\boldsymbol{s})) \frac{p(\boldsymbol{m}|\boldsymbol{s})}{p(\boldsymbol{m}|\tilde{\boldsymbol{s}})} p(\boldsymbol{m}|\tilde{\boldsymbol{s}}) \mathrm{d}\boldsymbol{m}$$

$$= \mathbb{E}_{p(\boldsymbol{m}|\tilde{\boldsymbol{s}})} \mathcal{L}(\boldsymbol{m}) \nabla_{\boldsymbol{s}} \log(p(\boldsymbol{m}|\boldsymbol{s})) \omega(\boldsymbol{m}), \tag{4}$$

where $\tilde{\boldsymbol{s}} \in \mathcal{S}$ is the outdated structural parameter. $\omega(\boldsymbol{m}) \triangleq \frac{p(\boldsymbol{m}|\boldsymbol{s})}{p(\boldsymbol{m}|\tilde{\boldsymbol{s}})}$ denotes the **reweighting function** that transforms the measure $p(\boldsymbol{m}|\boldsymbol{s})$ to $p(\boldsymbol{m}|\tilde{\boldsymbol{s}})$. Therefore,

$$\boldsymbol{g}_{\boldsymbol{s}} = \omega(\boldsymbol{m}) \mathcal{L}_{\mathcal{B}}(\boldsymbol{m}) \nabla_{\boldsymbol{s}} \log(p(\boldsymbol{m}|\boldsymbol{s})) \tag{5}$$

is an unbiased estimator of $\nabla_{\boldsymbol{s}} \mathbb{E}_{p(\boldsymbol{m}|\boldsymbol{s})} \mathcal{L}(\boldsymbol{m})$. $\boldsymbol{m}$ is sampled from the outdated distribution $p(\boldsymbol{m}|\tilde{\boldsymbol{s}})$.

**Discussion.** Let $\tilde{\boldsymbol{s}}$ and $\boldsymbol{s}$ be the outdated and current structural parameters, respectively. Eqn.(5) demonstrates that we can estimate the gradient of $\boldsymbol{s}$ in current iteration via reweigting the gradient estimated on an intermediate sub-model $\boldsymbol{m}$ instantiated at an earlier stage from an outdated distribution $p(\boldsymbol{m}|\tilde{\boldsymbol{s}})$. In this way, we can reduce the instantiation frequency. Moreover, the estimation of $\boldsymbol{g}_{\boldsymbol{s}}$ can be computationally and memory-efficient due to the compactess of sub-model $f(\cdot; \boldsymbol{W_m})$.

## 4.2 Framework of Efficient Pruning

**Variance Reduction.** Before giving the framework of our efficient pruning method, we need to address the issue of high variance in our reweighted policy gradient estimator in Eqn.(5) as follows:

(1) **Variance of $\omega(\boldsymbol{m})$:** The reweighting function $\omega(\boldsymbol{m})$ can be unstable due to large divergence between the two different distributions $p(\boldsymbol{m}|\boldsymbol{s})$ and $p(\boldsymbol{m}|\tilde{\boldsymbol{s}})$. We clip $\omega(\boldsymbol{m})$ to $[1 - \epsilon, 1]$ following PPO [42], and abbreviate it as $\mathrm{clip}(\omega(\boldsymbol{m}))$.

(2) **Variance of $\mathcal{L}_{\mathcal{B}}(\boldsymbol{m})$:** The training loss $\mathcal{L}_{\mathcal{B}}(\boldsymbol{m})$'s variation comes from the randomness of the mask $\boldsymbol{m}$. We minus $\mathcal{L}_{\mathcal{B}}(\boldsymbol{m}) \nabla_{\boldsymbol{s}} \log(p(\boldsymbol{m}|\boldsymbol{s}))$ by a highly correlated and several zero meaned items in form of $\mathcal{L}_{\mathcal{B}}(\boldsymbol{m}') \nabla_{\boldsymbol{s}} \log(p(\boldsymbol{m}|\boldsymbol{s}))$, where the mask $\boldsymbol{m}'$ is sampled independently with $\boldsymbol{m}$.

(3) **Variance of $\nabla_{\boldsymbol{s}} \log(p(\boldsymbol{m}|\boldsymbol{s}))$:** The potentially small denominator in $\nabla_{\boldsymbol{s}} \log(p(\boldsymbol{m}|\boldsymbol{s}))$, which takes the form of $\nabla_{\boldsymbol{s}} \log(p(\boldsymbol{m}|\boldsymbol{s})) = \frac{\boldsymbol{m}-\boldsymbol{s}}{\boldsymbol{s}(1-\boldsymbol{s})}$, could introduce additional variance. We multiply the gradient with a preconditioning matrix $H^{\alpha} = \mathrm{diag}(\boldsymbol{s} \circ (1 - \boldsymbol{s}))^{\alpha}$ with $\alpha \in [\frac{1}{2}, 1)$ to offset this impact.

After applying these variance reduction techniques, our reweighted policy gradient estimator becomes:

$$\boldsymbol{g}_{\boldsymbol{s}}^{vr} = \frac{1}{M - 1} \sum_{i=1}^{M} \mathrm{clip}(\omega(\boldsymbol{m}^{(i)})) \left( \mathcal{L}_{\mathcal{B}}(\boldsymbol{m}^{(i)}) - \frac{1}{M} \sum_{j=1}^{M} \mathcal{L}_{\mathcal{B}}(\boldsymbol{m}^{(j)}) \right) H^{\alpha}(\boldsymbol{s}) \nabla_{\boldsymbol{s}} \log p(\boldsymbol{m}^{(i)}|\boldsymbol{s}), \tag{6}$$

where the masks $\boldsymbol{m}^{(i)}, i = 1, \ldots, M$ are sampled independently from the distribution $p(\boldsymbol{m}|\tilde{\boldsymbol{s}})$.

We have the following theoretical guarantee for our variance reduced gradient $\boldsymbol{g}_{\boldsymbol{s}}^{vr}$. Detailed theoretical explanation is provided in the Appendix E.

**Theorem 1.** *Suppose the masks $\{\boldsymbol{m}^{(i)}\}_{i=1}^{M}$ are independently sampled from the distribution $p(\boldsymbol{m}|\tilde{\boldsymbol{s}})$, then for any $\alpha \in [\frac{1}{2}, 1)$ and $\boldsymbol{s} \in [0, 1]^{K}$, the variance of $\boldsymbol{g}_{\boldsymbol{s}}^{vr}$ is bounded.*

**Pruning Algorithm.** Finally, we present the complete algorithm in Algorithm 1. Based on reweighted policy gradient estimator with variance reduction, this algorithm enables efficient pruning using only the forward pass of the completely sparsified model, thereby avoiding the need for expensive backpropagation.

**Remark 1.** *Though we have $M$ sub-models, during implementation, they can be processed sequentially in order to save the memory cost. Concurrently, we assign distinct mini batches to each sub-model to ensure equitable allocation of computational resources with the baselines. See Appendix A.2 for details. Experiments under such fair setting verify the superiority of our approach.*

---

**Algorithm 1** Efficient Pruning with Reweighted Policy Gradient Estimator

---

**Input:** prune rate $\rho$, structural parameters $s$, instantiation steps $\tau$, dense DNN $W$, learning rate $\eta$.
 1: **Initialize $s$** with certain method.
 2: **for** $t = 1, 2, \ldots, T$ **do**
 3:     Set $\tilde{s} = s$, sample masks $\{m^{(i)}\}_{i=1}^{M}$ from $p(m|\tilde{s})$, instantiate sub-model $f(\cdot; W_{m^{(i)}})$.
 4:     **for** $r = 1, 2, \ldots, \tau$ **do**
 5:         Sample a mini batch $\mathcal{B} = \{(x_1, y_1), \ldots, (x_B, y_B)\}$.
 6:         Get $\mathcal{L}_{\mathcal{B}}(m^{(i)})$ from each sub-model $f(\cdot; W_{m^{(i)}})$ and compute $\omega(m^{(i)}) = \frac{p(m|s)}{p(m|\tilde{s})}$.
 7:         Calculate $g_s^{vr}$ according to Eqn.(6).
 8:         Update $s \leftarrow \text{proj}_{\mathcal{S}}(s - \eta g_s^{vr})$.
 9:     **end for**
10: **end for**
11: **return** Pruned DNN $W_m$, where mask $m$ is sampled from the distribution $p(m|s)$.

---

**Discussion.** We aummarize the appealing features of our algorithm as follows:

- Instantiating a compact sub-model loads only active components into memory, reducing memory usage and accelerating forward passes, unlike existing methods [57, 54, 58] that merely mask parameters with zeros and miss the full benefits of sparsity.

- If we consider training model weights through backpropagation:

$$W \leftarrow W - \eta_w g_w, \text{ where } g_w = \mathbb{E}_{p(m|\tilde{s})}\omega(m)\nabla_W \mathcal{L}(m) \approx \omega(m)\nabla_W \mathcal{L}_{\mathcal{B}}(m), \quad (7)$$

  the instantiation method ensures that gradient information maintains a sparsity level of $\rho$, granting an efficient backpropagation.

- We find that our algorithm is inherently well-suited for distributed sparse training. Through the use of reweighting technique, it reduces the overhead caused by frequent communication of model parameters in training process. Therefore, we present the application of our algorithm within the framework of distributed sparse training below.

### 4.3 Extension to Distributed Sparse Training

Distributed training requires the frequent passing of gradients and parameters between the server and clients, which imposes high demands on transfer efficiency. As discussed above, our algorithm is inherently well-suited for extending to distributed sparse training. During training process, the server updates the structural parameters based on policy gradient and the client is responsible for training its own sub-model, which only requires the latest structural information from the server. Since the weight parameters are trainable now, we use clearer notation $\mathcal{L}_{\mathcal{B}}(W_{m^{(i)}})$ instead of $\mathcal{L}_{\mathcal{B}}(m^{(i)})$ to distinguish this setting from the pruning task above. Our training process can be divided into four stages:

(1) **Instantiating sub-models.** To start the training round, the server samples $\{m^{(i)}\}_{i=1}^{M}$ from $p(m|s)$ to instantiate $\{f(\cdot; W_{m^{(i)}})\}_{i=1}^{M}$ and sends them to each client.

(2) **Updating sub-model paras.** At each iteration, client$_i$ receives reweighting value $\omega(m^{(i)})$ from the server, which provides real-time structural information and is used for updating the current parameters $W_{m^{(i)}}$ via Eqn.(7).

(3) **Updating structural paras.** Each client returns the loss $\mathcal{L}_{\mathcal{B}}(W_{m^{(i)}})$. By collecting these losses, the server estimates the policy gradient $g_s^{vr}$ via Eqn.(6) and updates $s$.

(4) **Aggregating updates.** Repeat (2) and (3) for $\tau$ steps. Then each client returns the parameter updates $\Delta W_{m^{(i)}}$ and the server aggregates them to update the full model.

Throughout the training process, **only two scalars**, the reweighting function $\omega(\boldsymbol{m}^{(i)})$ and the training loss $\mathcal{L}_{\mathcal{B}}(\boldsymbol{W}_{\boldsymbol{m}^{(i)}})$ of each client are communicated in real time, while the model weight parameters are passed only after every $\tau$ steps. Therefore, in this framework, the communication costs between the server and clients, as well as the instantiation overhead, are greatly reduced. Fig. 4 visually illustrates the flow of our algorithm. The detailed algorithm is shown in Appendix B.

**Discussion.** We genuinely believe that the distributed sparse training extension we propose is highly non-trivial and elegant. However, its advantages are challenging to fully convey through text alone. We strongly encourage readers to dedicate additional time to review its details along with Fig. 4 and Algorithm 2 in appendix.

# 5 Experiments

We conduct extensive experiments in this section, which demonstrate the superior performance of our algorithm. In Section 5.1, we introduce the experimental setups. In Section 5.2, we compare our method with other pruning methods for LLMs and conduct several ablation experiments in Section 5.3. Section 5.4 presents experiments using vision models in distributed sparse training scenario.

## 5.1 Experimental Setups

**Model Configuration.** We utilize the Llama and OPT families including Llama2-(7B, 13B) [48], Llama3\3.1-8B [12], Llama3.2-3B, and OPT-(2.7B, 6.7B) [64]. We apply our pruning algorithm across various structural granularities of the model, including attention heads and MLP channels.

**Prune Rate.** We target high sparsity levels (30%–50%) to better highlight our method's advantages. Under these settings, our instantiated method achieves higher acceleration and memory efficiency.

**Baseline.** Due to resource limits, we only include limited comparisons with optimization-based methods (e.g., Compresso [18], NutePrune [31]) in Section 5.3. Our main focus is on metric-based SOTA methods, including LLM-Pruner [34], SliceGPT [3], Wanda-sp, and FLAP [2]. No weight fine-tuning is performed post-pruning for fairness.

**Training Details.** We follow [34], using C4 [40] for training and Wikitext2 [36], PTB [35] for testing, which reflect the generalization of the pruned model. Adam [26] is used to optimize structural parameters $s$ with a learning rate of 5e-3 and batch size 8. Training is conducted for one epoch with frozen model weights. Further details are provided in Appendix A.

## 5.2 Pruning Results on LLMs

The main experimental results are presented in Table 1. At various sparsity levels, our pruning algorithm achieves lower perplexity compared to other state-of-the-art methods. This advantage is even more pronounced at 50% sparsity, highlighting the superior performance of our optimization-based algorithm over other metric-based methods.

Table 1: Performance comparison across methods and models under different prune rates. The lowest and second lowest PPL are in **bold** and underlined, respectively. - denotes unsupported model types.

| Prune rate | Method | Llama2-7B | | Llama2-13B | | Llama3-8B | | Llama3.1-8B | | Llama3.2-3B | | OPT-2.7B | | OPT-6.7B | |
|---|---|---|---|---|---|---|---|---|---|---|---|---|---|---|---|
| | | ↓WikiText2 | Ptb | WikiText2 | Ptb | WikiText2 | Ptb | WikiText2 | Ptb | WikiText2 | Ptb | WikiText2 | Ptb | WikiText2 | Ptb |
| 0% | Dense | 5.47 | 8.39 | 4.88 | 7.67 | 6.14 | 10.60 | 6.24 | 10.59 | 7.81 | 12.65 | 12.37 | 15.16 | 10.92 | 13.17 |
| 30% | LLM-Pruner | 27.13 | 111.16 | 15.19 | 125.96 | 20.18 | 30.37 | 19.23 | 31.50 | - | - | - | - | - | - |
| | SliceGPT | 23.31 | 88.74 | 17.85 | 72.38 | 50.88 | 75.37 | - | - | - | - | - | - | 19.70 | 25.49 |
| | Wanda-sp | 23.00 | 37.53 | 11.48 | 15.97 | 34.33 | 50.74 | 27.15 | 36.86 | 47.76 | 91.43 | 32.60 | 39.33 | 56.33 | 58.12 |
| | FLAP | 23.76 | 30.93 | 11.12 | 14.10 | 26.68 | 30.94 | 25.61 | 33.59 | 40.00 | 74.09 | 54.60 | 54.60 | 45.26 | 43.87 |
| | Ours | **10.80** | **15.85** | **9.75** | 15.24 | **17.78** | **29.50** | **17.89** | **25.88** | **22.71** | **39.61** | **21.05** | **25.39** | **19.47** | **23.85** |
| 40% | LLM-Pruner | 43.79 | 173.51 | 23.53 | 183.98 | 45.30 | 49.43 | 43.27 | 48.66 | - | - | - | - | - | - |
| | SliceGPT | 41.16 | 148.59 | 33.30 | 108.46 | 89.07 | 126.08 | - | - | - | - | - | - | 32.04 | 44.10 |
| | Wanda-sp | 39.57 | 71.72 | 24.63 | 50.08 | 74.57 | 128.06 | 61.62 | 90.66 | 76.47 | 137.37 | 70.11 | 74.63 | 148.41 | 108.58 |
| | FLAP | 48.12 | 81.70 | 19.01 | 27.37 | 70.63 | 84.47 | 54.72 | 68.52 | 126.06 | 185.33 | 90.02 | 87.25 | 102.00 | 95.82 |
| | Ours | **19.51** | **35.64** | **12.61** | **17.40** | **30.91** | **46.88** | **29.77** | **63.90** | **50.18** | **73.76** | **33.11** | **46.70** | **28.77** | **35.81** |
| 50% | LLM-Pruner | 178.32 | 424.44 | 48.94 | 269.79 | 129.16 | 200.27 | 141.25 | 153.84 | - | - | - | - | - | - |
| | SliceGPT | 71.72 | 237.90 | 53.14 | 161.87 | 155.70 | 186.02 | - | - | - | - | - | - | 60.57 | 89.41 |
| | Wanda-sp | 101.72 | 112.80 | 103.90 | 162.22 | 92.73 | 160.47 | 71.92 | 115.90 | 89.04 | 176.88 | 173.51 | 163.00 | 295.15 | 184.70 |
| | FLAP | 105.90 | 231.26 | 51.70 | 117.19 | 138.51 | 220.25 | 78.77 | 132.87 | 161.58 | 323.84 | 168.17 | 143.85 | 286.07 | 196.62 |
| | Ours | **26.83** | **44.23** | **21.21** | **44.47** | **57.14** | **74.13** | **57.80** | **78.35** | **83.88** | **116.41** | **65.86** | **90.02** | **53.75** | **65.86** |

We evaluate the generalization performance of our pruned models based on the settings from [34], using five zero-shot tasks from LM Evaluation Harness [16]: PIQA [5], HellaSwag [62], ARC-e, ARC-c [9], and OBQA [37], and recorded their average performance. We use Llama2-7B and Llama3-8B as baseline models. As shown in the Table 2, structured pruning does have a significant impact on the model's generalization ability. At high sparsity levels, datasets like ARC-e and ARC-c show a notable performance decline. However, our algorithm still maintains the best among all methods, demonstrating its effectiveness in preserving model capacity.

Table 2: Comparison of zero-shot performance between Llama2-7B and Llama3-8B.

| Prune rate | Method | Llama2-7B | | | | | | Llama3-8B | | | | | |
|---|---|---|---|---|---|---|---|---|---|---|---|---|---|
| | | ↑PIQA | HellaSwag | ARC-e | ARC-c | OBQA | Average | PIQA | HellaSwag | ARC-e | ARC-c | OBQA | Average |
| 0% | Dense | 78.74 | 76.00 | 74.62 | 46.33 | 44.20 | 63.98 | 80.79 | 79.19 | 77.69 | 53.41 | 45.00 | 67.22 |
| 30% | LLM-Pruner | 72.69 | 58.51 | 57.07 | 33.45 | 38.60 | 52.06 | 69.37 | 44.63 | 54.08 | 30.63 | 31.20 | 45.98 |
| | SliceGPT | 73.39 | 60.41 | 52.36 | 32.25 | 32.80 | 50.24 | 70.02 | 57.34 | 49.54 | 30.20 | 32.00 | 47.82 |
| | Ours | 73.61 | 63.06 | 58.46 | 34.90 | 38.80 | 53.76 | 72.74 | 58.58 | 56.52 | 30.89 | 33.60 | 50.47 |
| 40% | LM-Pruner | 67.19 | 45.22 | 43.39 | 27.30 | 31.80 | 42.98 | 63.55 | 36.28 | 41.96 | 23.63 | 27.60 | 38.60 |
| | SliceGPT | 66.81 | 48.98 | 41.88 | 25.94 | 27.00 | 42.12 | 64.20 | 44.54 | 37.63 | 24.49 | 28.00 | 39.77 |
| | Ours | 68.99 | 53.89 | 48.70 | 29.69 | 32.20 | 46.69 | 65.94 | 45.03 | 46.13 | 26.02 | 30.40 | 42.70 |

To demonstrate the effectiveness of our algorithm in reducing memory usage and accelerating the forward pass during the pruning process, we present the instantiated model parameter nums, multiply-accumulate counts (MAC), memory usage, and inference speed at different prune rates in Table 3. It is clear that the resource savings achieved by our algorithm are proportional to the model's sparsity.

Table 3: Statistics on pruned model. Inference Speed is evaluated by generating 400 tokens.

| Prune rate | Params(B) | MACs(G) | Memory(MiB) | Speed(×) |
|---|---|---|---|---|
| 0% | 6.61 | 845.71 | 12607.57 | 1.00 |
| 30% | 4.66 | 596.06 | 8888.24 | 1.24 |
| 35% | 4.32 | 552.57 | 8239.75 | 1.30 |
| 40% | 4.01 | 512.85 | 7648.47 | 1.43 |
| 45% | 3.67 | 469.90 | 6999.97 | 1.55 |
| 50% | 3.35 | 429.09 | 6389.62 | 1.58 |

Table 4: Comparison of initialization methods for $s$ on Llama2-7B, using PPL.

| Dataset | Prune rate | Random | $1 - \rho$ | Wanda-sp | FLAP |
|---|---|---|---|---|---|
| WikiText2 | 30% | 16.10 | 12.51 | 10.80 | 13.72 |
| | 40% | 32.81 | 22.30 | 19.51 | 18.10 |
| | 50% | 68.19 | 52.77 | 26.83 | 33.84 |
| | Average | 39.03 | 29.19 | **19.05** | 21.89 |
| Ptb | 30% | 24.31 | 22.18 | 15.85 | 18.10 |
| | 40% | 66.41 | 44.55 | 35.64 | 30.09 |
| | 50% | 117.33 | 73.51 | 44.23 | 61.22 |
| | Average | 69.35 | 46.75 | **31.91** | 36.47 |

## 5.3 Furthur Analysis

We conduct further studies on the following aspects: 1) different initialization methods for the structural parameters $s$; 2) effectiveness of the reweighting and instantiation technique; 3) ablation study for the variance reduction techniques; 4) comparison with the optimization-based methods.

**Initialization Methods Comparsion.** Four methods are employed to initialize the structural parameters $s$, including: 1) Uniform$(0, 1)$ random values; 2) $1 - \rho$ directly; 3) the Wanda-sp score; 4) the FLAP score[1]. From the results in Table 4, it can be observed that using Wanda-sp initialization yields the best performance, followed closely by FLAP initialization. This indicates that initializing the structural parameters $s$ with heuristic metrics can facilitate the optimization process. Additionally, we find that random initialization is already sufficient to surpass metric-based pruning algorithms.

**Effects of Reweighting and Instantiation.** As shown in Fig. 2, as the number of instantiation steps $\tau$ increases, perplexity gradually rises due to the accumulation of errors from the outdated sub-model $W_m$ and gradients $g_s$. However, by using the reweighting function, this issue can be effectively alleviated. Even with 100 steps, the perplexity on WikiText2 still reaches 71.01, which is better than SliceGPT's 71.72. Meanwhile, in (c), we observe that after applying reweighting, all models in the experiment exhibit a significant reduction in perplexity. Therefore, we believe the reweighting function makes infrequent instantiation possible, which helps us fully leverage the sparse structure, resulting in memory savings and forward acceleration.

---

[1]Since the scores of Wanda-SP and FLAP do not fall within the range $(0, 1)$, we normalize these scores to form a probability distribution. For different prune rates, we scale them to the desired sparsity level.

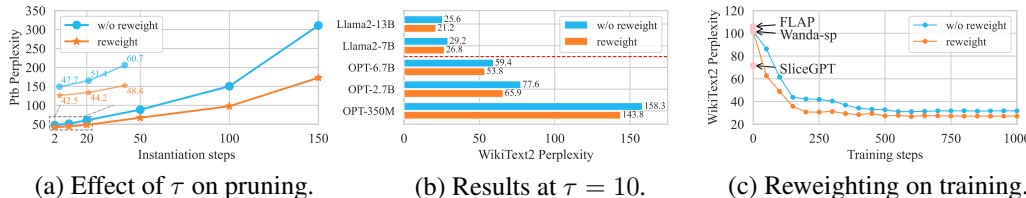

| (a) Effect of $\tau$ on pruning. | (b) Results at $\tau = 10$. | (c) Reweighting on training. |

Figure 2: (a) For different instantiation step $\tau$, reweighted version consistently outperforms the w/o reweighted baseline. Considering the trade-off between instantiation time and pruning performance, we fix $\tau = 10$ throughout all experiments. (b) Under our fixed setting $\tau = 10$ for all experiments, reweighting technique improves pruning across all models. Notably, on OPT-2.7B, the impact of reweighting can reach up to **11.7** compared to w/o reweighting case. (c) Reweighting technique stabilizes the training process, leading to better pruning. (a), (b), (c) are based on the 50% prune rate.

**Variance Reduction.** We employ clipping and preconditioning matrix $H$ to stabilize the training process and conducted ablation experiments to verify their contribution. The results in Table 5 shows that omitting either the clipping or the preconditioning matrix negatively impacts the performance of the pruned model. Among these, the clipping has a more evident effect on performance. Compared to our final algorithm, not using clipping increases the perplexity on the Wikitext2 dataset by 12.44, indicating that our variance reduction techniques enhance the stability of the training process. Details of the variance reduction analysis can be found in Appendix C.

Table 5: Variance reduction study, 50% Llama2-7B.

| Method | ↓ WikiText2 | Ptb | ↑ PIQA | HellaSwag | ARC-e | ARC-c | Average |
|--------|-------------|-----|--------|-----------|-------|-------|---------|
| Ours | **26.83** | **44.23** | **62.89** | **40.19** | **39.98** | **25.00** | **42.02** |
| w/o $H$ | 32.75 | 57.01 | 61.48 | 38.21 | 37.29 | 24.40 | 40.35 |
| w/o *clip* | 39.27 | 69.28 | 60.83 | 39.11 | 38.68 | 23.04 | 40.42 |

Note: evaluation metrics: Perplexity (WikiText2, PTB) and zero-shot accuracy (PIQA, HellaSwag, ARC-e, ARC-c and their average).

Table 6: Overhead test, 50% Llama2-7B.

| Method | WikiText2 | Forward(s) | Backward(s) | Memory(MiB) |
|--------|-----------|------------|-------------|-------------|
| Ours | 26.83 | 0.015 | None | 6390.21 |
| Compresso[†] | 49.28 | 0.024 | 0.041 | 33298.74 |
| NutePrune[‡] | 24.89 | 0.049 | 0.051 | 37460.78 |

Note: † pruning with LoRA adaption. ‡ co-training the masks and LoRA modules, also involves distillation.

**Comparison with optimization-based methods.** Our method outperforms other optimization-based pruning algorithms by significantly reducing memory usage and training time. To demonstrate the superiority of our algorithm, we select Compresso [18] and NutePrune [31] for comparison. Since these methods use different training settings, we focus on memory usage and time consumption in a single optimization iteration. Table 6 shows that our algorithm achieves comparable performance while consuming less memory, offering faster forward time, and requiring no backpropagation.

### 5.4 Distributed Sparse Training

Based on the method described in Section 4.3, we conduct extensive experiments within the framework of distributed sparse training. We use *gloo* [43] as the distributed backend and built an 8-node cluster for training on the ImageNet-1K dataset [41] (each node equipped with a GPU). The worker nodes are connected via a 40 Gbps (5000 Mb/s) Ethernet interface. The baseline algorithms include LocalSGD [45], PowerSGD [50], TSNNS [10], and our base solver EffTrain [67] (which performs sparse training locally without using reweighting techniques, described in Section 3). We set the instantiation sampling interval $\tau$ to 20 steps. The detailed experimental configuration and baseline description can be found in Appendix A. The specific comparison results are shown in Table 7.

Table 7: Comparison of methods across ResNet-50, MobileNet-V1 and DeiT-Base on ImageNet-1K.

| Method | ResNet-50 | | | | MobileNet-V1 [24] | | | | DeiT-Base [47] | | | |
|--------|-----------|-----------|------------|-------|--------|----------|----------|------|--------|----------|----------|------|
| | Acc(%)↑ | Params(%)↓ | FLOPs(%)↓ | Time↓ | Acc(%) | Params(%) | FLOPs(%) | Time | Acc(%) | Params(%) | FLOPs(%) | Time |
| Dense Dist. | 76.9 | 100 | 100 | 23.00 | 72.0 | 100 | 100 | 8.66 | 81.8 | 100 | 100 | 53.05 |
| Local SGD | 75.0 | 100 | 100 | 20.52 | _71.5_ | 100 | 100 | 6.15 | _80.6_ | 100 | 100 | 40.41 |
| PowerSGD | _76.0_ | 100 | 100 | _19.25_ | 71.3 | 100 | 100 | _5.50_ | 79.5 | 100 | 100 | _38.28_ |
| TSNNS | 74.5 | 63.2 | 77.4 | 19.49 | 70.4 | 88.5 | 82.9 | 5.69 | 78.5 | 80.3 | 79.4 | 39.37 |
| EffTrain | **76.1** | _48.2_ | _46.8_ | 115.04 | 71.5 | _68.1_ | _69.2_ | 61.36 | 80.1 | _67.1_ | _68.2_ | 288.24 |
| Ours | 75.2 | **48.1** | **46.5** | **9.79** | **71.6** | **67.4** | **68.9** | **4.15** | **80.9** | **66.5** | **67.3** | **26.70** |

Our method achieves excellent performance across three different models, surpassing other distributed training algorithms, while also reducing model params and FLOPs. The base solver EffTrain also

achieves superior performance and model compression; however, due to the frequent instantiation operations, its training time is significantly longer compared to distributed algorithms. In conclusion, our algorithm integrates well into the framework of distributed sparse training. By using the reweighting technique to reduce instantiation frequency, it achieves promising results in less time.

# 6 Conclusion

This paper introduces a novel DNN pruning algorithm that is computationally and memory-efficient. For the first time in pruning research, we define model instantiation to leverage sparsity for memory savings and forward acceleration. Additionally, we propose a reweighting technique that reduces instantiation frequency, making it feasible to prune LLMs under resource constraints. As a byproduct, our algorithm seamlessly integrates into distributed sparse training environments. Extensive experiments demonstrate the excellent practical performance of our pruning algorithm, and numerous ablation studies validates our proposed techniques' effectiveness. Besides, we also provide thorough theoretical analysis to support our algorithm.

# 7 Acknowledgements

This work was supported by the National Nature Science Foundation of China (62472097), Shanghai Municipal Science and Technology Commission (Grant No.24511106102), AI for Science Foundation of Fudan University (FudanX24AI028) and Fudan Kunpeng&Ascend Center of Cultivation. The computations in this research were performed on the CFFF platform of Fudan University.

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
