# OpenReview forum: "Computation and Memory-Efficient Model Compression with  Gradient Reweighting"
_NeurIPS.cc/2025/Conference — NeurIPS 2025 poster_

### Official Review · Reviewer_Hb8r · 2025-06-06

**Clarity:** 2
**Significance:** 3
**Originality:** 3
**Rating:** 4
**Confidence:** 3

**Summary:**

This paper proposes an optimization-based pruning method for large language models (LLMs). The key contribution is a reweighting technique that allows the reuse of outdated sub-models to estimate gradients for structural parameters, reducing the need to frequently instantiate new sub-networks. The method also incorporates variance reduction techniques to stabilize training. Experimental results demonstrate memory savings, inference speedup, and competitive performance compared to state-of-the-art pruning methods. The paper also extends the method to a distributed sparse training setting.

**Questions:**

Please refer to Strengths and Weaknesses

**Ethical Concerns:**

["NO or VERY MINOR ethics concerns only"]

**Limitations:**

The authors have not discussed the limitations and potential negative societal impact of their work. It is recommanded to be included in the final revision.

**Quality:**

3

**Strengths And Weaknesses:**

**Strengths**

1. The method allows using an outdated sub-model for multiple pruning steps, which significantly reduces computational cost.

2. The method achieves competitive or superior performance compared to several baselines in terms of perplexity, memory usage, and inference speed.

3. The paper is well organized and easy to follow.

**Weaknesses**

1. The introduction states that "Consequently, LLMs pose a unique challenge for pruning", but this claim lacks sufficient justification. A clearer explanation should be provided.

2. The theoretical parts in Section 4 are sometimes dense and hard to follow without a strong background in knowledge.

3. Comparisons with more recent optimization-based methods are limited. Most baselines are metric-based.

4. typos: caption of Figure 1 ("vanila" -> "vanilla")

---

> ### Author Rebuttal · Authors · 2025-07-31
>
> ## W1: Explanation for "LLMs pose a unique challenge for pruning".
>
> As we mentioned in the main text, unlike conventional DNNs, LLMs face unique challenges in pruning, primarily due to the asymmetry in resources (data volume and memory) between training and pruning. Conventional DNNs, e.g., ResNet for image classification, can typically be pruned under resource conditions similar to those used during training, such as comparable datasets and memory (e.g., ImageNet dataset and 4 GeForce RTX 4090), which is always affordable for the downstream practioners. In contrast, training LLMs often relies on extremely large-scale computational resources and massive data. For example, LLaMA3-405B was trained with up to 16K H100 GPUs and approximately 15T of training tokens [r1], which far exceeds what is accessible during the pruning phase. This characteristic of LLMs therefore presents unique challenges for pruning methods, distinct from those encountered in conventional DNNs.
>
> ## W2: Theoretical parts in Section 4 are dense and hard to follow.
>
> We will move some of the theoretical parts to the appendix in the revision.
>
> ## W3: Comparison with optimization-based baselines is limited.
>
> Thanks for your suggestion. We additionally include experiments based on the optimization-based pruning method Compresso, NutePrune, LoRAShear [r2], MaskPrune [r3] with different models and prune rates. More comparison will be included in the revised version if accepted. For a fair comparison under equal training time, we extend the training epochs of our method to fully exploit its effienciy. We additionally report the time and memory overhead incurred by each method during the pruning process. The results are summarized below.
>
> **Table R1. Comparison with various optimization-based methods on Llama2-7B. To ensure a consistent training time budget, we extend the training epoch accordingly. Meanwhile, we also record the memory overhead for reference. The fine-tuned version is trained on the Alpaca dataset.**
> |Prune rate|Method|WikiText2 PPL↓|Ptb PPL↓|WikiText2 PPL(&Fine-tune)↓|Ptb PPL(&Fine-tune)↓|Time cost|Memory(MiB)|
> |-|-|:-:|:-:|:-:|:-:|:-:|:-:|
> |0%|-|5.47|8.39|-|-|-|-|
> |20%|Ours(1 Epoch)|9.67|13.94|8.35|13.18|4.8h|10224.34|
> |20%|Ours(2 Epochs)|9.44|13.70|8.02|12.90|9.6h|10224.34|
> |20%|Ours(3 Epochs)|**9.35**|**13.48**|**7.94**|**12.57**|14.5h|10224.34|
> |20%|Compresso|11.47|21.25|9.92|18.39|16.8h|33298.74|
> |20%|NutePrune|9.88|16.64|8.74|14.90|13.4h|37460.78|
> |50%|Ours(1 Epoch)|26.83|44.23|14.39|30.07|3.6h|6390.21|
> |50%|Ours(2 Epochs)|23.46|41.79|12.79|25.42|7.2h|6390.21|
> |50%|Ours(3 Epochs)|**21.52**|**38.69**|**12.55**|**25.20**|10.7h|6390.21|
> |50%|Compresso|49.28|78.90|40.83|67.08|16.8h|33298.74|
> |50%|NutePrune|24.89|40.83|12.94|26.04|13.4h|37460.78|
>
> **Table R2. Comparison with various optimization-based methods on Llama-7B. Zero Shot task includes BoolQ, PIQA, HellaSwag, Winogrande, ARC-e, ARC-c and OBQA.**
> |Prune rate|Method|Zero Shot Avg. Score↑|Zero Shot Avg. Score(&Fine-tune)↑|Time cost|Memory(MiB)|
> |-|-|:-:|:-:|:-:|:-:|
> |0%|-|63.25|-|-|-|
> |20%|Ours(1 Epoch)|60.37|61.71|4.8h|10173.46|
> |20%|Ours(2 Epochs)|**61.34**|**61.97**|9.6h|10173.46|
> |20%|Compresso|-|60.75|16.8h|33298.74|
> |20%|NutePrune|59.46|61.46|13.4h|37460.78|
> |20%|LoRAShear|-|60.63|-|-|
> |20%|MaskPrune|61.17|-|-|-|
> |50%|Ours(1 Epoch)|51.73|53.26|3.6h|6239.06|
> |50%|Ours(2 Epochs)|**51.94**|**53.68**|7.2h|6239.06|
> |50%|Compresso|-|46.87|16.8h|33298.74|
> |50%|NutePrune|50.35|52.13|13.4h|37460.78|
> |50%|LoRAShear|-|50.39|-|-|
> |50%|MaskPrune|49.92|-|-|-|
>
> ## W4: Typos.
>
> We will correct these typos in the revision.
>
> ## L5: Missing discussion of limitations and broader impacts.
>
> Thanks for the reminder. In fact, we discussed the limitations and broader impacts in Appendices F and G. We will move them into main text in the revised version if accepted.
>
> ## Reference
>
> [r1] Dubey A, Jauhri A, Pandey A, et al. The llama 3 herd of models. arXiv e-prints, 2024: arXiv: 2407.21783.
>
> [r2] Chen T, Ding T, Yadav B, et al. Lorashear: Efficient large language model structured pruning and knowledge recovery. ArXiv 2023.
>
> [r3] Qin J, Tan J, Zhang K, et al. MaskPrune: Mask-based LLM Pruning for Layer-wise Uniform Structures. ArXiv 2025.

---

> > ### Author Response · Authors · 2025-08-09
> >
> > Dear Reviewer Hb8r,
> >
> > We sincerely appreciate for your support and have carefully responded to your constructive comments thoroughly.
> >
> > Since the deadline for the author-reviewer discussion phase is approaching, would you mind kindly letting us know if any further discussion or clarification is needed? Thank you again for your valuable time.

---

### Official Review · Reviewer_ubKW · 2025-06-22

**Clarity:** 3
**Significance:** 3
**Originality:** 3
**Rating:** 4
**Confidence:** 3

**Summary:**

This paper introduces a computationally and memory-efficient pruning algorithm for LLMs, which addresses a key practical challenge in optimization-based structured pruning — the high overhead of sub-model instantiation. The core idea is a reweighting scheme, which reuses the old sub-model gradients to estimate the current gradient. The paper integrates multiple variance reduction techniques and demonstrates the method’s advantages in both standalone and distributed pruning scenarios.

**Questions:**

See weaknesses.

If the authors can solve some of my confusion in experiments, the practical significance of the paper will be greatly improved, and I will be happy to increase my score.

**Ethical Concerns:**

["NO or VERY MINOR ethics concerns only"]

**Final Justification:**

The authors have addressed my concerns on the higher sparse level, different results in the original paper, comparison with the results with fine-tuning, and statistical information. I think those improvements increase the value of the paper, and I have changed the rating from 2 to 4.

**Limitations:**

Yes

**Quality:**

3

**Strengths And Weaknesses:**

Strengths
1. Novel reweighting-based pruning technique with strong theoretical and practical grounding.
2. Extensive experimental coverage across LLaMA, OPT, and vision models.

Weaknesses
1. I'm wondering about the results beyond 50% sparsity and robustness under extreme pruning, which is lacking in the paper.
2. SliceGPT reported that the WikiText2 of Llama2-7B at 30% sparsity is 8.12, which significantly differs from the one reported in this paper (23.31). The WikiText2 of Llama2-13B at 30% sparsity is also larger than the one in the original paper. The difference also occurs in the results of LLM-pruner, I'm wondering why there is such a gap. Is there some difference in the experimental setup? If so why not use the same setup as the original paper?
3. Comparison with optimization-based baselines is limited. Only 50% sparsity is compared, more sparsity levels should be compared. Nuteprune reported that the WikiText2 of Llama2-7B at 50% sparsity is 12.29, which significantly differs from the one reported in this paper. Does this difference come from the lack of fine-tuning?
4. Since the other methods do not appear to match the results presented in their own original papers. The quality of these results in a scientific context is low, because it is difficult to draw any conclusions from them. I hope the author can provide some explanation for the difference.
5. I want to know why the authors only compare the results without fine-tuning? If the models pruned by gradient reweighting can't beat those SOTA methods after fine-tuning, the practical significance of gradient reweighting would seem to be less. I recommend that the authors compare the performance after fine-tuning.
6. I wish the authors could provide statistical information rather than single experiments, which would help to increase the credibility of the results

---

> ### Author Rebuttal · Authors · 2025-07-31
>
> ## W1: Results beyond 50% sparsity.
>
> We provide experimental results under 55% and 60% prune rates, as shown in the Table R1.
>
> **Table R1. Performances with higher prune rates on  Llama2-7B.**
> |Prune rate|Method|WikiText2 PPL↓|Ptb PPL↓|
> |-|-|:-:|:-:|
> |55%|LLM-Pruner|434.51|531.02|
> |55%|SliceGPT|97.69|311.01|
> |55%|Wanda-sp|218.80|164.81|
> |55%|FLAP|363.82|413.05|
> |55%|Ours|**35.34**|**59.83**|
> |60%|LLM-Pruner|761.97|816.07|
> |60%|SliceGPT|139.64|475.73|
> |60%|Wanda-sp|298.84|247.60|
> |60%|FLAP|464.39|532.67|
> |60%|Ours|**53.84**|**72.95**|
>
> ## W2: The experimental performance gap in SliceGPT and LLM-Pruner.
>
> The reason of the performance difference is we use different   experimental setup for fair comparison.
>
> Before presenting our detailed setting, we'd like to clarify the following three factors:
>
> 1. **SliceGPT uses WikiText2 as both the calibration and evaluation dataset, which makes it difficult to fully evaluate the generalization ability of the pruned model.**  Such experimental setup is also not wildely adopted in the literature.
> 2. Regarding the performance difference on LLM-Pruner, there may be a misunderstanding: the original paper used the LLaMA-1 with a evaluation context length of 128, whereas our experiments are based on the LLaMA-2 and LLaMA-3 with a context length of 2048.
> 3. **Pruning methods such as SliceGPT and LLM-Pruner can have notable performance variation over different calibration datastes and input sequence lengths.** As shown in Table R2 below, it achieves an average score of 51.50 when using WikiText2 as the validation set, compared to 57.93 when using Alpaca. Figure 7 of SliceGPT further illustrates that when varying the length of the calibration dataset, the WikiText2 PPL of OPT-6.7B fluctuates within a range of 12.0 to nearly 20.0.
>
> **Table R2. The results extracted from Tables 7–10 in the original SliceGPT paper demonstrate that the performance of LLaMA2-7B at a 30% prune rate varies significantly depending on the calibration data used. "-" means the result is not available in the original paper.**
> |Calibration Data|WikiText2 PPL↓|PIQA↑|WinoGrande↑|HellaSwag↑|ARC-e↑|ARC-c↑|Avg. Score↑|
> |-|:-:|:-:|:-:|:-:|:-:|:-:|:-:|
> |WikiText2|8.12|63.55|61.33|49.62|51.77|31.23|51.50|
> |WikiText2&fine-tuning|-|67.41|63.22|55.65|50.76|34.13|54.23|
> |Alpaca|-|72.25|59.83|55.86|63.93|37.80|57.93|
> |Alpaca&fine-tuning|-|74.59|61.64|63.06|66.54|40.87|61.34|
>
> **Our experimental settings.** To ensure fair comparison across all baselines in our experiments, we standardize the calibration dataset to **C4** and use a sequence length of **128**. For evaluation, we use **WikiText2** and **Ptb** as test sets with a length of **2048**, and we **do not apply fine-tuning**.
>
> **Results under the same setting with SliceGPT.** Furthermore, we evaluate our method under the same experimental setting as SliceGPT, where **WikiText2 is used as both the training and evaluation set**. Under this experimental setup, our method also surpasses SliceGPT on multiple models. The results are shown in the following table.
>
> **Table R3. Performance of our method, with training and evaluation conducted on Wikitext2(↓).**
> |Prune rate|Method|Llama2-7B|Llama2-13B|OPT-2.7B|OPT-6.7B|
> |-|-|:-:|:-:|:-:|:-:|
> |20%|SliceGPT|6.64|5.81|13.73|11.48|
> |20%|Ours|**6.32**|**5.63**|**12.69**|**10.94**|
> |30%|SliceGPT|8.12|6.99|15.83|12.51|
> |30%|Ours|**7.75**|**6.61**|**15.02**|**12.07**|
>
> ## W3-1: Comparison with optimization-based baselines is limited.
>
> Thanks for your suggestion. We additionally include experiments based on the optimization-based pruning method Compresso, NutePrune, LoRAShear [r1], MaskPrune [r2] with different models and prune rates. More comparison will be included in the revised version if accepted. For a fair comparison under equal training time, we extend the training epochs of our method to fully exploit its effienciy. We additionally report the time and memory overhead incurred by each method during the pruning process. The results are summarized below.
>
> **Table R4. Comparison with various optimization-based methods on Llama2-7B. To ensure a consistent training time budget, we extend the training epoch accordingly. Meanwhile, we also record the memory overhead for reference. The fine-tuned version is trained on the Alpaca dataset.**
> |Prune rate|Method|WikiText2 PPL↓|Ptb PPL↓|WikiText2 PPL(&Fine-tune)↓|Ptb PPL(&Fine-tune)↓|Time cost|Memory(MiB)|
> |-|-|:-:|:-:|:-:|:-:|:-:|:-:|
> |0%|-|5.47|8.39|-|-|-|-|
> |20%|Ours(1 Epoch)|9.67|13.94|8.35|13.18|4.8h|10224.34|
> |20%|Ours(2 Epochs)|9.44|13.70|8.02|12.90|9.6h|10224.34|
> |20%|Ours(3 Epochs)|**9.35**|**13.48**|**7.94**|**12.57**|14.5h|10224.34|
> |20%|Compresso|11.47|21.25|9.92|18.39|16.8h|33298.74|
> |20%|NutePrune|9.88|16.64|8.74|14.90|13.4h|37460.78|
> |50%|Ours(1 Epoch)|26.83|44.23|14.39|30.07|3.6h|6390.21|
> |50%|Ours(2 Epochs)|23.46|41.79|12.79|25.42|7.2h|6390.21|
> |50%|Ours(3 Epochs)|**21.52**|**38.69**|**12.55**|**25.20**|10.7h|6390.21|
> |50%|Compresso|49.28|78.90|40.83|67.08|16.8h|33298.74|
> |50%|NutePrune|24.89|40.83|12.94|26.04|13.4h|37460.78|
>
> **Table R5. Comparison with various optimization-based methods on Llama-7B. Zero Shot task includes BoolQ, PIQA, HellaSwag, Winogrande, ARC-e, ARC-c and OBQA.**
> |Prune rate|Method|Zero Shot Avg. Score↑|Zero Shot Avg. Score(&Fine-tune)↑|Time cost|Memory(MiB)|
> |-|-|:-:|:-:|:-:|:-:|
> |0%|-|63.25|-|-|-|
> |20%|Ours(1 Epoch)|60.37|61.71|4.8h|10173.46|
> |20%|Ours(2 Epochs)|**61.34**|**61.97**|9.6h|10173.46|
> |20%|Compresso|-|60.75|16.8h|33298.74|
> |20%|NutePrune|59.46|61.46|13.4h|37460.78|
> |20%|LoRAShear|-|60.63|-|-|
> |20%|MaskPrune|61.17|-|-|-|
> |50%|Ours(1 Epoch)|51.73|53.26|3.6h|6239.06|
> |50%|Ours(2 Epochs)|**51.94**|**53.68**|7.2h|6239.06|
> |50%|Compresso|-|46.87|16.8h|33298.74|
> |50%|NutePrune|50.35|52.13|13.4h|37460.78|
> |50%|LoRAShear|-|50.39|-|-|
> |50%|MaskPrune|49.92|-|-|-|
>
> ## W3-2: Performance differences on NutePrune. Does this difference come from the lack of fine-tuning?
>
> Yes. To ensure a fair comparison, we did not perform additional fine-tuning when reproducing the experiments of NutePrune, which may lead to discrepancies from the results reported in the original paper.
>
> ## W4: Explanation for the difference.
>
> For the differences in certain experimental results, please refer to the responses in W2 and W3-2.
>
> ## W5: Why the authors only compare the results without fine-tuning?
>
> Thanks for your suggestions. We provide the fine-tuning results in Table R4 and Table R5. For fair comparison, we adopt the same setup as LLM-Pruner: LoRA fine-tuning is performed on the pruned model using the Alpaca dataset for 2 epochs, with a rank of 8 and a learning rate of 1e-4.
>
> ## W6: Statistical information.
>
> Thank you for your valuable suggestion. Actually, we observed that the experimental results exhibited low variance in the initial training stage, so we fixed a random seed for all subsequent experiments. To illustrate this, we conducted statistical experiments using five different random seeds on both Llama2-7B and Llama3-8B. The results, shown in the table below, indicate that the standard deviation is relatively small compared to the performance gains. Complete results with statistical information will be included in the revised version if accepted.
>
> **Table R6. Statistical information over 5 random seeds.**
> |Model|WikiText2 PPL↓|Ptb PPL↓|PIQA↑|HellaSwag↑|ARC-e↑|ARC-c↑|OBQA↑|
> |-|:-:|:-:|:-:|:-:|:-:|:-:|:-:|
> |Llama2-7B|10.58(±0.64)|15.86(±0.73)|73.59(±1.24)|63.09(±0.93)|58.51(±1.53)|32.85(±0.87)|38.62(±0.91)|
> |Llama3-8B|17.88(±0.79)|29.26(±0.96)|72.18(±1.66)|58.68(±0.85)|56.50(±0.56)|30.87(±0.75)|33.68(±0.64)|
>
> ## Reference
>
> [r1] Chen T, Ding T, Yadav B, et al. Lorashear: Efficient large language model structured pruning and knowledge recovery. ArXiv 2023.
>
> [r2] Qin J, Tan J, Zhang K, et al. MaskPrune: Mask-based LLM Pruning for Layer-wise Uniform Structures. ArXiv 2025.

---

> ### Comment · Reviewer_ubKW · 2025-08-03
>
> I thank the authors for their detailed response. The explanations and additional experiments are very helpful and have addressed my concerns. I'm very happy to increase the rating.

---

> > ### Author Response · Authors · 2025-08-03
> >
> > We sincerely appreciate your follow-up response and are grateful for your recognition of our efforts. If you have any further questions or suggestions, we would be happy to continue the discussion. Thank you again for your thoughtful review.

---

### Official Review · Reviewer_bDVm · 2025-07-02

**Clarity:** 3
**Significance:** 2
**Originality:** 3
**Rating:** 4
**Confidence:** 1

**Summary:**

This paper addresses the challenges of pruning large language models (LLMs), where conventional methods struggle due to the high computational and memory overhead of repeated sub-model instantiations. The authors propose a novel, resource-efficient pruning approach that introduces a reweighting mechanism to estimate gradients of the current pruned model using gradients from previously instantiated sub-models. This significantly reduces the frequency of model instantiation. Additional techniques such as gradient clipping and preconditioning are introduced to stabilize optimization and reduce variance. The method is empirically validated across domains, achieving 50\% sparsity and 1.58$\times$ forward-pass speedup on LLaMA2-7B using only 6 GB of memory, with improved perplexity and zero-shot performance compared to prior work. Furthermore, the technique demonstrates strong potential for distributed sparse training, achieving 2$\times$ speedup over dense baselines.

**Questions:**

Is the m in Equation 1 either 0 or 1? Can we prune part of the each sub-module?

**Ethical Concerns:**

["NO or VERY MINOR ethics concerns only"]

**Final Justification:**

The rebuttal addresses my confusion and makes the paper solid. I will raise my score to 4.

**Limitations:**

See above.

**Paper Formatting Concerns:**

No formatting issues.

**Quality:**

3

**Strengths And Weaknesses:**

Strengths:
1. The paper presents a clear and concise mathematical formulation, including only the essential equations—particularly for the variance reduction mechanism—without unnecessary complexity.
2. The ablation studies are thorough and well-structured, effectively isolating the impact of each proposed component on addressing key bottlenecks in pruning.
3. The experimental evaluation is comprehensive, covering both algorithmic metrics (e.g., perplexity, zero-shot performance) and practical system-level outcomes (e.g., memory usage and actual speedup).

Weaknesses:
1. The manuscript repeatedly emphasizes the three main contributions throughout multiple sections, leading to redundancy and detracting from the overall clarity and flow of the writing.
2. The empirical comparison with prior optimization-based pruning methods is limited, relying on only two baseline works. A broader and more diverse set of baselines would strengthen the validity of the claims.
3. The trade-off introduced by the variance reduction techniques appears to slightly degrade model performance in Table 5. It would be helpful to evaluate whether variance reduction is strictly necessary—can the method maintain acceptable accuracy while further improving throughput and memory consumption if the variance reduction is omitted?

---

> ### Author Rebuttal · Authors · 2025-07-31
>
> ## W1: Repeated emphasis on three main contributions.
>
> We will revise the manuscript to improve clarity and make it easier for readers to follow.
>
> ## W2: Lack of comparison with optimization-based methods.
>
> Thanks for your suggestion. We additionally include experiments based on the optimization-based pruning method Compresso, NutePrune, LoRAShear [r1], MaskPrune [r2] with different models and prune rates. More comparison will be included in the revised version if accepted. For a fair comparison under equal training time, we extend the training epochs of our method to fully exploit its effienciy. We additionally report the time and memory overhead incurred by each method during the pruning process. The results are summarized in **Table R1** and **Table R2**.
>
> **Table R1. Comparison with various optimization-based methods on Llama2-7B. To ensure a consistent training time budget, we extend the training epoch accordingly. Meanwhile, we also record the memory overhead for reference. The fine-tuned version is trained on the Alpaca dataset.**
> |Prune rate|Method|WikiText2 PPL↓|Ptb PPL↓|WikiText2 PPL(&Fine-tune)↓|Ptb PPL(&Fine-tune)↓|Time cost|Memory(MiB)|
> |-|-|:-:|:-:|:-:|:-:|:-:|:-:|
> |0%|-|5.47|8.39|-|-|-|-|
> |20%|Ours(1 Epoch)|9.67|13.94|8.35|13.18|4.8h|10224.34|
> |20%|Ours(2 Epochs)|9.44|13.70|8.02|12.90|9.6h|10224.34|
> |20%|Ours(3 Epochs)|**9.35**|**13.48**|**7.94**|**12.57**|14.5h|10224.34|
> |20%|Compresso|11.47|21.25|9.92|18.39|16.8h|33298.74|
> |20%|NutePrune|9.88|16.64|8.74|14.90|13.4h|37460.78|
> |50%|Ours(1 Epoch)|26.83|44.23|14.39|30.07|3.6h|6390.21|
> |50%|Ours(2 Epochs)|23.46|41.79|12.79|25.42|7.2h|6390.21|
> |50%|Ours(3 Epochs)|**21.52**|**38.69**|**12.55**|**25.20**|10.7h|6390.21|
> |50%|Compresso|49.28|78.90|40.83|67.08|16.8h|33298.74|
> |50%|NutePrune|24.89|40.83|12.94|26.04|13.4h|37460.78|
>
> **Table R2. Comparison with various optimization-based methods on Llama-7B. Zero Shot task includes BoolQ, PIQA, HellaSwag, Winogrande, ARC-e, ARC-c and OBQA.**
> |Prune rate|Method|Zero Shot Avg. Score↑|Zero Shot Avg. Score(&Fine-tune)↑|Time cost|Memory(MiB)|
> |-|-|:-:|:-:|:-:|:-:|
> |0%|-|63.25|-|-|-|
> |20%|Ours(1 Epoch)|60.37|61.71|4.8h|10173.46|
> |20%|Ours(2 Epochs)|**61.34**|**61.97**|9.6h|10173.46|
> |20%|Compresso|-|60.75|16.8h|33298.74|
> |20%|NutePrune|59.46|61.46|13.4h|37460.78|
> |20%|LoRAShear|-|60.63|-|-|
> |20%|MaskPrune|61.17|-|-|-|
> |50%|Ours(1 Epoch)|51.73|53.26|3.6h|6239.06|
> |50%|Ours(2 Epochs)|**51.94**|**53.68**|7.2h|6239.06|
> |50%|Compresso|-|46.87|16.8h|33298.74|
> |50%|NutePrune|50.35|52.13|13.4h|37460.78|
> |50%|LoRAShear|-|50.39|-|-|
> |50%|MaskPrune|49.92|-|-|-|
>
> ## W3: The variance reduction in Table 5 seems to degrade the performance.
>
> There may be some misunderstanding. As shown in Table 5, removing the components of our variance reduction technique (either the clipping operation or the preconditioning matrix H) leads to a performance degrade. "Ours" in Table 5 refers to our pruning method equipped with the full variance reduction technique. It is important to note that for WikiText2 and Ptb, the evaluation metric is perplexity, where lower values indicate better performance, whereas for PIQA, HellaSwag, ARC-e, and ARC-c, the metric is accuracy, where higher values are better. Taken together, these results demonstrate that the variance reduction strategies contribute positively to performance and remain an integral part of our method.
>
> ## Q1: Is the $m$ in Equation 1 either 0 or 1? Can we prune part of the each sub-module?
>
> First, yes, in Equation 1, $m$ is either 0 or 1, representing the retention state of a sub-module. Then, we can prune part of each sub-module by setting finer-grained mask. In existing works (e.g., Sheared LLaMA [r3] and FLAP [r4]), structural pruning is typically applied at the level of attention heads or the intermediate dimensions of FFNs, as such sparsity patterns can be efficiently implemented on various hardware platforms to achieve speedup. Therefore, our method adopts these default settings. However, thanks to the flexibility of our masks, our approach can also be extended to support finer-grained pruning. For instance, within each attention head, we can assign individual masks to the intermediate dimensions corresponding to Q, K, and V, enabling more fine-grained pruning strategies.
>
> ## Reference
>
> [r1] Chen T, Ding T, Yadav B, et al. Lorashear: Efficient large language model structured pruning and knowledge recovery. ArXiv 2023.
>
> [r2] Qin J, Tan J, Zhang K, et al. MaskPrune: Mask-based LLM Pruning for Layer-wise Uniform Structures. ArXiv 2025.
>
> [r3] Xia M, Gao T, Zeng Z, et al. Sheared LLaMA: Accelerating Language Model Pre-training via Structured Pruning. ICLR 2024.
>
> [r4] An Y, Zhao X, Yu T, et al. Fluctuation-based adaptive structured pruning for large language models. AAAI 2024.

---

> > ### Comment · Reviewer_bDVm · 2025-08-03
> >
> > Thanks for the detailed response. I will raise my score.

---

> > > ### Author Response · Authors · 2025-08-04
> > >
> > > We sincerely appreciate your response and your willingness to raise the score. If there are any additional questions or suggestions, we would be more than happy to discuss them.

---

### Official Review · Reviewer_FimZ · 2025-07-03

**Clarity:** 4
**Significance:** 3
**Originality:** 3
**Rating:** 4
**Confidence:** 5

**Summary:**

This work introduces a reweighting-based mask updating method for pruning DNNs, particularly Large Language Models (LLMs), with a focus on computational and memory efficiency. The primary challenge addressed is the computational complexity and memory redundancy associated with instantiating intermediate pruned sub-models. Instead of relying on back-propagation for the entire neural network, this method estimates the gradient of the pruned network. The authors highlight that the DNN feed-forward-only mechanism can also be generalized to sparse distributed training. The results demonstrate state-of-the-art performance in LLM and DNN compression compared to existing pruning algorithms.

**Questions:**

Super LLMs and Pruning Limitations
* Resource Requirements: What are the potential resource requirements for super LLMs to demonstrate the limitations of pruning?
* Network Growth and Redundancy: Theoretically, at what point does network growth indicate sufficient redundancy to warrant pruning?

Distributed Sparse Training and Asynchronous Pruning
* Full-Sync Optimization: Does the current distributed sparse training framework necessitate full-synchronization for optimization?
* Asynchronous Training and Framework Adjustments: If asynchronous training is implemented, what adjustments are required? Would a separate framework be necessary to accommodate asynchronous pruning?

**Ethical Concerns:**

["NO or VERY MINOR ethics concerns only"]

**Final Justification:**

All of my concerns has been addressed by the rebuttal. After combining all the comments, I'd like to keep the rating 4 for this work.

**Limitations:**

yes

**Quality:**

3

**Strengths And Weaknesses:**

Strengths
* The reweighted policy gradient estimator approximates the gradient of the mask distribution parameter using outdated sub-model instantiation results, significantly reducing model instantiation frequency.
* Variance reduction in the pruning framework is mathematically proven to be bounded, demonstrating the framework's stability for practical application.
* The framework runs in parallel with model training, making it compatible with existing methods for simultaneous model finetuning and compression.
* As a byproduct, the framework is also compatible with sparse distributed training.
* The reweighting-based pruning framework achieves state-of-the-art performance compared to existing pruning frameworks for DNNs and LLMs.
* The presentation is logical, smooth, and clear, making it easy for future researchers to extend the work. Detailed discussions on theorems, implementation, and code are provided in the appendix.




Weakness
* The study highlights that pruning performance on Large Language Models (LLMs) remains largely unexplored. While the proposed method outperforms other LLM pruning techniques, the pruned network significantly underperforms the original LLM. This disparity might stem from LLMs' inherent sensitivity to noise. However, computational resource limitations prevented the application of larger, more redundant networks to validate this hypothesis. Consequently, further research is needed to determine if the trade-off between LLM complexity and efficiency justifies this approach to LLM pruning.
* minor: line 159 compactess -> compactness

---

> ### Author Rebuttal · Authors · 2025-07-31
>
> ## W1: Further research is needed to determine if the trade-off between LLM complexity and efficiency justifies this approach to LLM pruning.
>
> We'd like to clarify as follows.
>
> **Reasons on Unexplored Pruning for LLMs.** We observe that LLMs often exhibit performance degradation after pruning, especially under high sparsity structural pruning. We argue this phenomenon may be attributed to two main factors:
>
> 1. A mismatch between pruning and training resources. The data scale and computing resources used for pruning are much lower than those used for training, which hinders its ability to achieve the desired effectiveness.
> 2. Potentially lower redundancy in LLMs compared to vision models. One possible explanation is that language data contains less redundancy than image data, in which local pixel regions typically exhibit high similarity. However, this remains unclear under the current limited pruning resources.
>
> Based on the above background, we emphasis the key contributions of our work as follows:
>
> 1. The memory cost during pruning is significantly reduced by our method, making it more feasible to prune larger models with the same memory resource.  This is important for the downstream practioners of LLMs, where the resource is always limited. As shown in **Table R1**, our method incurs very low memory overhead.
>
> 2. Our method has higher effiency. We emphasize that improving pruning performance  is closely tied to the efficiency of the algorithms, especially for the large scaled networks like LLMs. A more efficient pruning algorithm allows for more iterative updates, enabling more sufficient exploration and refinement on sparse structures. As shown in **Table R1**, extending the training time to match that of the baselines leads to more significant performance superority.
>
> 3. Our pruning method achieves SOTA performance compared to the baselines.
>
> **Table R1. Comparison of pruning performance, training time, and memory overhead under 50% prune rate of Llama2-7B. To ensure fair experimentation, we extended the training schedule of our method to match comparable time budgets.**
> |Method|WikiText2 PPL↓|Ptb PPL↓|Time cost|Memory(MiB)|
> |-|:-:|:-:|:-:|:-:|
> |Ours(1 Epoch)|26.83|44.23|3.6h|6390.21|
> |Ours(2 Epochs)|23.46|41.79|7.2h|6390.21|
> |Ours(3 Epochs)|**21.52**|**38.69**|**10.7h**|**6390.21**|
> |Compresso|49.28|78.90|16.8h|33298.74|
> |NutePrune|24.89|40.83|13.4h|37460.78|
>
> ## Q1: What are the potential resource requirements for super LLMs to demonstrate the limitations of pruning?
>
> For medium-sized networks, i.e., ResNet for image classification, existing pruning algorithms often require resource investments comparable to full training in order to achieve satisfactory pruning results, like 4 GeForce RTX 4090. By analogy, for LLMs with more complex parameter structures and larger parameter scales, it is reasonable to hypothesize that approaching the limit of pruning performance similarly demands computational resources and data volumes on par with pre-training.
>
> ## Q2: Theoretically, at what scale does network growth imply sufficient redundancy to make pruning worthwhile?
>
> Most existing neural network inherently contain a certain degree of redundancy, as they are manually designed and cannot guarantee that each parameter is optimally utilized to represent knowledge. Consequently, pruning is essential for improving efficiency in applications where extreme time efficiency is required.
>
> ## Q3: Does the current distributed sparse training framework necessitate full-synchronization for optimization?
>
> Our framework can be integrated with both  synchronized and asynchronized optimization flexibly. It is currently designed as a synchronized optimization process, since synchronized optimization is widely used in distribued training systems.
>
> To support asynchronous optimization, only Step 8 in Algorithm 2 needs to be modified. Instead of waiting for all workers to return their parameter updates $\Delta W_{m(i)}$ before updating the global model, the server can update the global weights using each received gradient individually as follows:
>
> For each incoming $\Delta W_{m(i)}$ from worker i, update the global weights as:
>
> $$\qquad\qquad\qquad\qquad\qquad\qquad W \leftarrow W + \eta_{3} \Delta W_{m(i)},$$
>
> where $\eta_{3}$ is a small learning rate that controls the pace of asynchronous updates.
>
> This minor modification would allow the framework to run fully asynchronously without requiring a complete redesign, while still benefiting from the lightweight communication scheme enabled by the sparse structure and reweighting strategy.
>
> ## Q4: If asynchronous training is implemented, what adjustments are required? Would a separate framework be necessary to accommodate asynchronous pruning?
>
> The asynchronous version can be implemented following the answer in **Q3**, without the need for a separate training framework. We provide experimental results of asynchronous sparse training on ResNet-50, as shown in **Table R2**.
>
> **Table R2. Comparison of asynchronous sparse training results**
> |Method|Acc(%)↑|Params(%)↓|FLOPs(%)↓|Time↓|
> |-|:-:|:-:|:-:|:-:|
> |Dense Dist.|76.9|100|100|23.00|
> |Local SGD|75.0|100|100|20.52|
> |PowerSGD|76.0|100|100|19.25|
> |TSNNS|74.5|63.2|77.4|19.49|
> |EffTrain|76.1|48.2|46.8|115.04|
> |Ours(Syn)|75.2|48.1|46.5|9.79|
> |Ours(Asyn)|74.7|48.2|46.7|9.13|

---

> > ### Comment · Reviewer_FimZ · 2025-08-08
> >
> > Thanks for addressing the concerns in the review and detailed explanation. I have no other concerns.

---

> > > ### Author Response · Authors · 2025-08-09
> > >
> > > We sincerely appreciate your feedback and recognition, and are pleased that our work has addressed your concerns.

---

### Note · Authors · 2025-08-14

Dear ACs and Reviewers,

We sincerely thank you for your thorough engagement. We would like to provide the following concise summary of the current scoring status and our key contributions.

**Current status**

Based on the Reviewers’ feedbacks during the rebuttal phase, the current scores could be positive. Reviewer bDVm and ubKW explicitly stated in their responses that their concerns had been addressed and that they would raise their scores. Reviewers FimZ and Hb8r initially gave a score of 4, and FimZ has confirmed that his concerns were resolved with no further issues.

**Key contributions supported by the reviewers' positive feedback**
1. We propose a novel reweighting-based pruning algorithm that effectively **reduces the frequency of model instantiations, improves computational efficiency, and lowers memory usage** (Reviewers FimZ and Hb8r). We think our method is particularly valuable in memory-constrained scenarios, enabling more thorough optimization of large model pruning under given computational resources.

2. Our proposed reweighting-based pruning algorithm is novel with **strong theoretical and practical grounding**, such as bounded variance, demonstrating its stability and reliability (Reviewers FimZ and ubKW).

3. **Comprehensive experiments with competitive or superior performance** (Reviewers FimZ, bDVm, ubKW, and Hb8r). Our experiments cover a variety of models (e.g., LLaMA, OPT, vision models) and multiple evaluation metrics (including perplexity, zero-shot performance, memory usage, and speedup), achieving or surpassing existing methods on several key indicators.

In the rebuttal, we mainly provided additional comparison experiments with optimization-based pruning methods and presented more detailed settinng in the reproduced experiments involving SliceGPT, LLM-Pruner, and NutePrune. The corresponding Reviewers FimZ, bDVm, and ubKW confirmed their concerns were resolved. **Notably, Reviewers bDVm and ubKW stated they will raise their scores.** The initial ratings of the other two Reviewers are positive.

Once again, we sincerely thank ACs and Reviewers for their careful evaluation and valuable feedback throughout this process.

Yours sincerely,

Authors

---

### Decision · Program_Chairs · 2025-09-17

**Decision:**

Accept (poster)

**Comment:**

This manuscript proposes a method to explore pruned LLMs based on having access to a dense fully trained LLM.  Their approach is novel in that they are able to better make use of the potential pruned intermediate LLMs with less variance in the reweight function, the loss, and the gradient.  These are believed to allow for a more stable training process when searching for the final pruned LLM.  The reviewers initial manuscript was viewed by the reviewers as needing additional experiments, especially needing studies with fine-tuning training.  The authors engaged well with these criticism and the result is that all four reviewers gave scores of 4 with the only variation being on the certainty of the reviewers in their scores; that said the feedback from the less certain reviewers was consistent with that of the more certain reviewers.

The manuscript is a reasonable contribution.  Using variance reduction for these less stable quantities is a natural and good idea.  It has been implemented and shown to give superior performance.  Reviewers have the requisite expertise to review the results and the only remaining question is how well the methods scale to larger problems which is beyond the scope of what the authors would be able to do.  The ideas of the manuscript could be conveyed well through a poster; there isn't need for an oral or spotlight presentation.